# An Experimental Determination of Critical Power for Self-Focusing of Femtosecond Pulses in Air Using Focal-Spot Measurements

Huiting Song [1], Zuoqiang Hao [1,2,3], Bingxin Yan [1], Faqian Liu [1], Dongwei Li [1,2,3], Junwei Chang [1], Yangjian Cai [1,2,3] and Lanzhi Zhang [1,2,3,*]

[1] Shandong Provincial Engineering and Technical Center of Light Manipulations & Shandong Provincial Key Laboratory of Optics and Photonic Device, School of Physics and Electronics, Shandong Normal University, Jinan 250014, China; zqhao@sdnu.edu.cn (Z.H.); lidongwei@sdnu.edu.cn (D.L.); yangjiancai@sdnu.edu.cn (Y.C.)

[2] Collaborative Innovation Center of Light Manipulation and Applications, Shandong Normal University, Jinan 250358, China

[3] Joint Research Center of Light Manipulation Science and Photonic Integrated Chip of East China Normal University and Shandong Normal University, East China Normal University, Shanghai 200241, China

* Correspondence: lzzhang@sdnu.edu.cn

**Abstract:** The filamentation of femtosecond pulses has attracted significant attention, owing to its unique characteristics and related applications. The self-focusing critical power of femtosecond pulses is one of the key parameters in the filamentation process and its application. However, the experimental determination of this power remains a challenging task. In this study, we propose an experimental approach to investigating the critical power for self-focusing of both femtosecond Gaussian and vortex beams with relatively low topological charges by analyzing the changes in the focal spot at different propagation distances. Our work offers a practical and convenient method for determining the self-focusing critical power of femtosecond pulses.

**Keywords:** critical power for self-focusing; focal spot; focus shifting; vortex beam

## 1. Introduction

Intense femtosecond laser pulses can propagate over very long distances in the form of filamentation with a clamped intensity [1–3]. This unique phenomenon has attracted significant interest among researchers due to its diverse range of potential applications in various fields, including supercontinuum and third harmonic generation [4–6], pulse self-compression [7–9], remote sensing [10,11], air waveguides [12–14], air lasing [15–17], laser-guided lightning [18,19], filament-induced breakdown spectroscopy [11,20–22], combustion diagnostics and laser ignition [23,24]. It is well known that this nonlinear phenomenon is the result of dynamic balance mainly between the optical Kerr self-focusing effect and the plasma defocusing effect. When the incident laser power exceeds a certain value, the self-focusing effect can overcome the diffraction of the beam, resulting in the collapse of the beam. This specific value is known as the critical power for self-focusing, denoted as $P_{cr}$ given by $P_{cr} = 3.77\lambda^2/8\pi n_2 n_0$, where $\lambda$ is the laser central wavelength, $n_0$ denotes the linear index of refraction, and $n_2$ is the coefficient of the nonlinear index of refraction. It is a fundamental parameter in studies of filamentation, and is very useful in various applications. However, this formula is given under a continuous-wave (cw) laser condition. It has been demonstrated that many factors including beam profile, group velocity dispersion, pulse duration, and external focusing conditions have significant influences on the critical power [25–30]. It seems unfeasible, in practice, to directly determine the exact value of nominal critical power for self-focusing of ultrashort laser pulses [29]. For the same reason, the nominal critical power calculated using the aforementioned formula

may not be suitable for ultrashort laser pulse cases. This leads to a lack of practical reference values of the critical power for self-focusing of ultrashort laser pulses, and thus it is necessary to determine quantitatively the critical power for different conditions. Liu et al. directly determined the critical power for self-focusing of femtosecond laser pulses in air by measuring the movement of the focus with the increase in the input laser energy [27]. By using this focus-shifting method, the critical powers in helium [31] and flame [32] were further successfully obtained. Akturk et al. proposed a P-scan method to evaluate different propagation regimes of femtosecond laser pulses, and determine the critical powers for filamentation and multi-filamentation regimes [33]. Kudryashov et al. determined the critical power for self-focusing by observing the difference between the luminous track and Rayleigh lengths (initiation of filamentation) [30]. Recently, by using a pinhole located at a focus region, we obtained experimentally and numerically the critical power for femtosecond filamentation in air [34]. Li and Zhang et al. explored the threshold of filamentation in liquid by using interferometry [35,36]. For structured laser beams, the situation becomes more complicated. The critical power for self-focusing of vortex beams was initially derived by Kruglov et al. [37]. Subsequently, Fibich et al. achieved a simplified expression for this critical power [38]. In our recent experimental study, we proposed a simple method to determine the critical powers for self-focusing of both femtosecond Gaussian and vortex beams in air by measuring fluorescence emission with a photomultiplier tube [39].

As pointed out by Liu et al. in Ref. [27], the most direct way for determining the critical power is to analyze the change in laser intensity along the laser propagation direction [40,41]. However, it is extremely challenging to accurately measure the intensity, especially at the focal region, mainly due to the change of the pulse shape during the self-focusing process and the high intensity involved. Therefore, using phenomena closely associated with the high intensity, such as ablation, diffraction, fluorescence or acoustic emission, becomes a more practical approach to investigate the laser filamentation. Among these methods, the ablation spot method is widely used for characterizing laser filamentation [42,43]. Recently, Nishibata et al. conducted a study on the focusing properties of ultrashort laser pulses in laser processing by analyzing the irradiation and ablation areas [44]. Their findings indicate that the ablation technique can effectively demonstrate the evolution of intensity or fluence in the laser focal region. Therefore, the ablation method could be a promising approach for determining the critical power for self-focusing of femtosecond laser pulses. In this study, we propose a simple and efficient experimental method to measure the critical power. By analyzing the ablation region on photosensitive paper at various laser propagation distances and input laser energies, we determine the critical power for self-focusing of femtosecond Gaussian and vortex beams. Additionally, our findings reveal that the proposed method is not applicable to vortex beams with a higher topological charge. Nevertheless, we demonstrate that the ablation technique remains a practical and valuable approach for studying the dynamics of laser self-focusing and determining the critical power for self-focusing.

## 2. Experiment Setup

The schematic of our experimental setup to measure the self-focusing critical power is shown in Figure 1. The laser source was an amplified Ti:sapphire femtosecond laser system (Solstice Ace, Spectra-Physics, Milpitas, CA, USA) with a central wavelength of 790 nm, pulse duration of 65 fs, beam diameter of 9.5 mm ($1/e^2$), a repetition rate of 1 kHz, and maximum pulse energy of 4.2 mJ. However, our experiment was carried out under the conditions of relatively low laser power, typically within the range of several GW, and a relatively large beam diameter. The potential influence of the focusing lens and other optical elements used in the experiment on the beam quality is considered to be negligible. A combination of a half-wave plate (HWP20-800B, LBTEK, Changsha, China) and a polarizer (WP25L-UB, Thorlabs, Newtown, NJ, USA) was used to adjust the laser energy. Subsequently, a vortex plate (VRx-800, LBTEK) and two quarter-wave plates (QWP20-800B. LBTEK) were employed to generate vortex beams. In order to control the

exposure time, a shutter was positioned after the vortex beam generation system, set to a duration of 30 ms. A lens with a focal length of 500 mm was used to focus the femtosecond laser pulse in air. The pulse duration after optics elements used in our experiment was optimized by adjusting the compressor of the laser system through pre-compensating for the pulse dispersion introduced by the elements. The photosensitive paper, affixed to a translation stage, captured the focal spot generated by 30 pulses, which were controlled by a mechanical shutter (GCI-7103M, Daheng Optics, Beijing, China). Finally, the ablation patterns were recorded using a microscope (alpha300 R, WITec, Ulm, Germany).

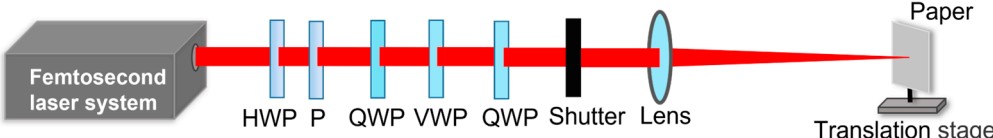

**Figure 1.** The schematic of the experimental setup. HWP: half-wave plate. P: polarizer. QWP: quarter-wave plate. VWP: vortex wave plate.

## 3. Results and Discussion

Figure 2 shows several typical patterns ablated by femtosecond Gaussian and vortex beams with different topological charges (1, 2, and 3, respectively). These patterns are obtained by placing photosensitive paper before the focal point, enabling a clear visualization of the singularities in the center of the vortex beams. We can see from the patterns that the ablation area primarily corresponds to the region with high laser intensity, which is a commonly used method to investigate the evolution of femtosecond filamentation [45]. To investigate the evolution of the ablation in the focal region, we observed the ablation patterns at different positions using the photosensitive paper. As an example, we can take the case of vortex beams, and the ablation patterns at various positions around the focal region of the beam are presented in Figure 3. The patterns clearly show the evolution of the vortex beams around the focus. The ablation area undergoes a reduction as the beam approaches the focal position, whereas it experiences an expansion upon departure. Note that the singularities of vortex beams near the focus are too small to be recognized from the ablation patterns. The peak intensities are evaluated based on photographic paper measurements, as an example, at distances of 498 mm and 494 mm for input laser energies of 250 μJ and 550 μJ, respectively. These values are estimated to be 2.22 TW/cm$^2$ and 2.26 TW/cm$^2$. However, it is expected that these calculated intensities would be lower when compared to those using metal samples, due to the fact that the beam size is overestimated in our experiment. The explanation will be provided in the following section. Furthermore, the change in the ablation area serves as an indicator of the laser intensity, which enables the exploration of the evolution of the laser beam along its propagation direction [44]. Therefore, by systematically analyzing the variation of the ablation area in relation to the propagation distance as the input energy gradually increases, we can explore the evolution of the laser intensity and, subsequently, determine the critical power for self-focusing of femtosecond pulses.

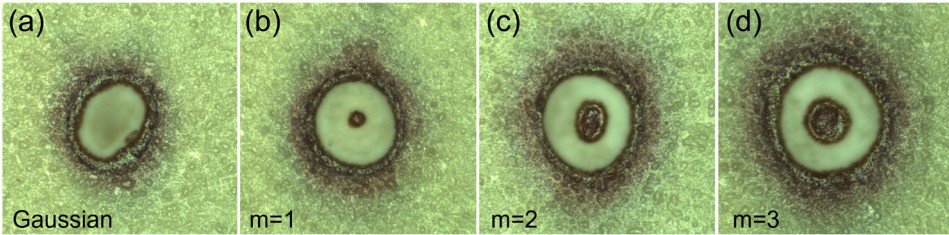

**Figure 2.** Typical ablation patterns by femtosecond (**a**) Gaussian and vortex beams with topological charge (**b**) *m* = 1, (**c**) *m* = 2, and (**d**) *m* = 3, respectively.

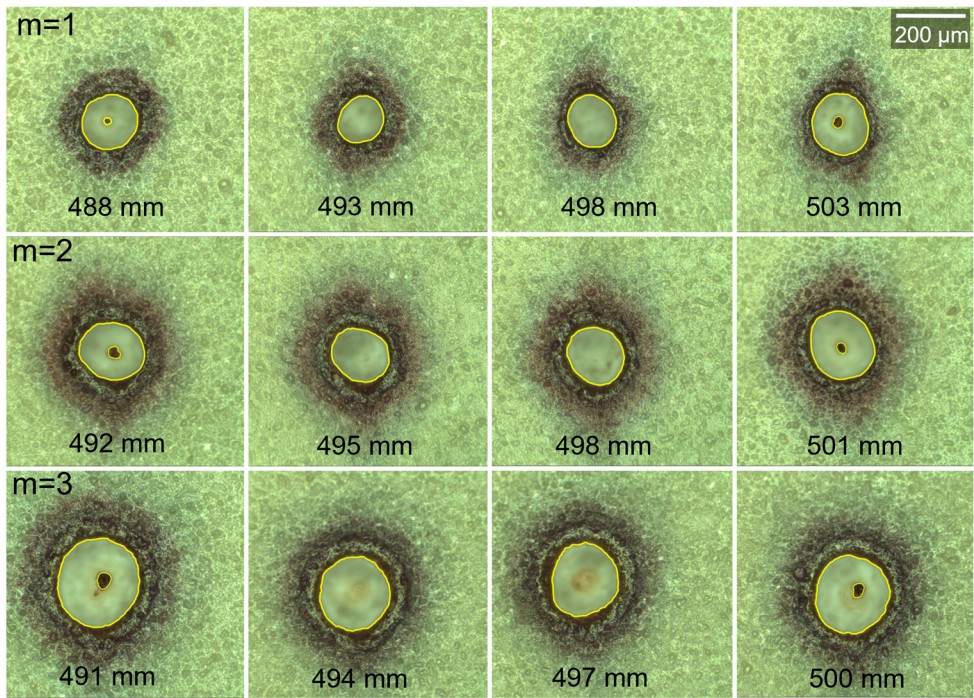

**Figure 3.** Ablation patterns of femtosecond vortex beam with *m* = 1 (250 μJ), *m* = 2 (350 μJ), and *m* = 3 (550 μJ) at several typical propagation distances.

First, we evaluated the ablation patterns of the Gaussian beam, and then quantified the areas of the ablated regions using ImageJ software (Ver. 1.8) [46]. During the analysis, we maintained the same tolerance of 30 in the Wand Tool of the software. The Wand Tool of ImageJ software allowed us to select a contiguous area under the condition such that all pixel values in that area were within the range of the initial value ± tolerance. The ablation area can be automatically and properly selected by using this tool. For instance, the selected areas for the vortex beam patterns are marked by yellow lines in Figure 3. Subsequently, the area of the region of interest was calculated using the software. The results are plotted in Figure 4a, with the data being fitted using the polynomial fitting method. It is evident that for each input laser energy, there exists a minimum ablation area. Moreover, as the input energy increases, the position corresponding to the minimum area undergoes a shift towards the focusing lens, illustrating the manifestation of the self-focusing effect. Here, the displacement of the minimum area serves as an indicator of the movement of the beam focus. Hence, the determination of the critical power for self-focusing through the ablation method is also based on the measuring of the focus-shifting. Therefore, we followed the approach of the focus-shifting method [27]. The focus (minimum area) positions for various input energies are plotted in Figure 4b. The evolution of the position with a minimum area as the function of input energy exhibits the same trend as those obtained using the methods of focus-shifting and fluorescence measurement [27,39]. The position experiences negligible change when the input energy is relatively low, and then undergoes a rapid shift with further increases in input lase energy. To identify the deviation position, linear fittings were applied to the data. The intersection of the two red fitted lines can be considered the critical power point, from which the critical power for self-focusing of the femtosecond Gaussian beam can be obtained. The calculated critical power was approximately 2.05 GW. The value was very close to that obtained in our previous work by using fluorescence measurements [39].

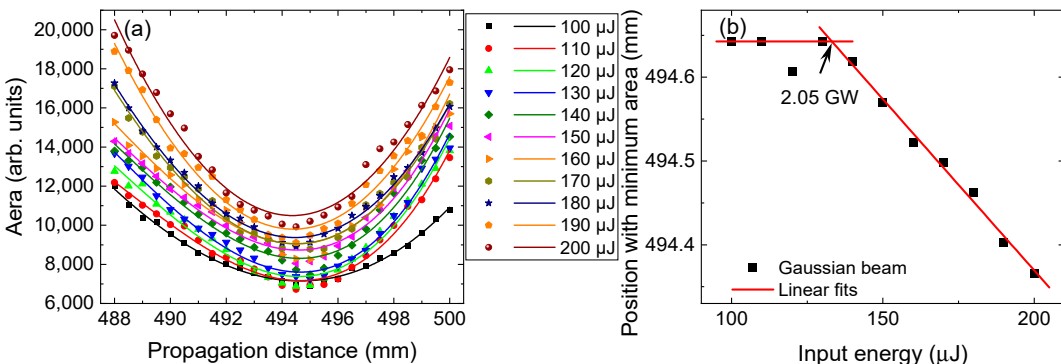

**Figure 4.** (**a**) The areas of ablated spots by Gaussian beam as a function of propagation distance for various input laser energies. Corresponding fitted lines are also plotted in the figure. (**b**) The change in the position with a minimum area of the ablated spot as the increase in the input energy. Linear fitting is used to find the deviation position, which can be considered the critical power point.

We then applied the method to the case of vortex laser beams. The results are shown in Figure 5. It is evident that the areas of the ablated spots by vortex beams with $m = 1$ and 2 have a similar evolution with the increasing input laser energy to that of Gaussian beam. By fitting the experimental data using polynomial fitting method, the positions with minimum areas were obtained and are plotted in Figure 5b,d for the two cases, respectively. A similar trend to the Gaussian case was observed. Hence, the same linear fitting method was used to find the deviation positions, indicated by the arrows in Figure 5b,d, and the critical powers for the self-focusing of the two vortex beams were successfully obtained, which were found to be 3.48 GW and 6.12 GW, respectively. These values are in good agreement with the values obtained in our recent study, where the critical powers for the self-focusing of vortex beams were determined through fluorescence measurements [39].

We extended the study to include a vortex beam with a higher topological charge $m = 3$. The areas of ablated spots were evaluated and are plotted in Figure 6. We can see that the areas start to show irregular evolutions with the increase in input energy, compared to the above three cases. Especially for higher input energies, the data have much bigger fluctuations along the propagation direction. The polynomial fitting lines are not reliable any more. The $R^2$ of the fitting becomes worse with the increase in input energy, and it is only 66% for 900 µJ case. Under this condition, it is hard to find a position which has the minimum area. This phenomenon is similar to what we observed in our previous work [39], where the fluorescence peak position of air ionization became worse with the increase in the topological charge (as can be seen from Figure 6 in the Ref. [39]). The main reason for this should be that the non-uniform distribution of laser intensity in the ring-shape cross-section of vortex beam influences greatly the focus-shifting process [39]. On the other hand, the coincidence is not surprising, because both the ablation method proposed in this study and the fluorescence measurements obtained using CCD imaging used in Ref. [39] are basically the measuring of the focus shifting. Therefore, it can be concluded that the method is not applicable for vortex beams with higher topological charges either.

We noticed that the area in the focal region looked more diffused in the case of vortex beams than that in the Gaussian beam case. A possible mechanism is as follows. The distinct intensity distribution of the Gaussian and vortex beams leads to different outcomes under the influence of self-focusing. For the Gaussian beam, the laser energy concentrates gradually to the beam axis. However, for the vortex beam, the ring-shape distributed laser energy will be concentrated into a narrower ring. Consequently, any non-uniformity in the intensity distribution of the ring-shaped beam will be amplified in the subsequent self-focusing process, intensifying energy competition and promoting modulation insta-bility. Consequently, vortex beams are expected to exhibit greater fluctuations in focus size compared to Gaussian beams as the incident laser energy increases. On the other hand, in our experiments, each ablation pattern was generated by approximately 30 laser

pulses, which was determined by the limitations of our equipment. This accumulation will unavoidably introduce deviations to the ablation pattern due to the drift of filamentation in air [47], thereby reducing the precision of the spot area evolution along the propagation distance. To improve the accuracy of this method in determining the critical power, it is highly recommended to employ single pulse ablation, followed by statistical analysis.

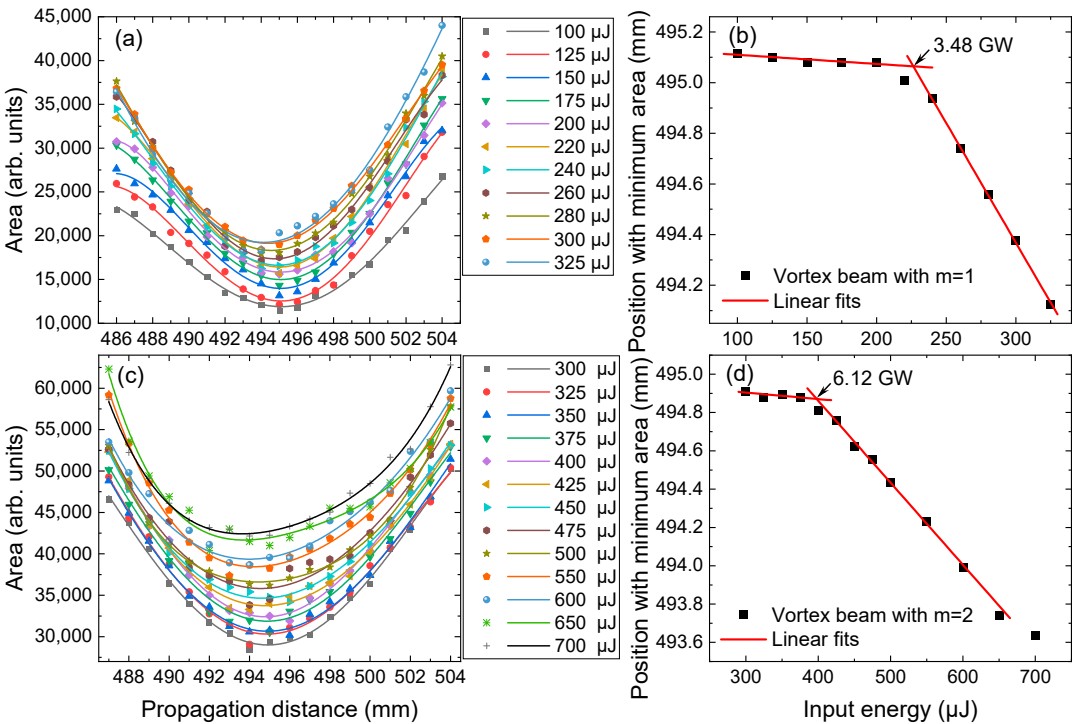

**Figure 5.** The areas of the ablated spots by vortex beams with (**a**) *m* = 1 and (**c**) *m* = 2 as functions of propagation distance for various input laser energies. Fitted lines are plotted to find the positions with minimum areas. (**b**,**d**) plot the positions with minimum areas of the ablated spots as the increase in the input energy. Linear fitting is used to find the deviation position which, can be considered the critical power point.

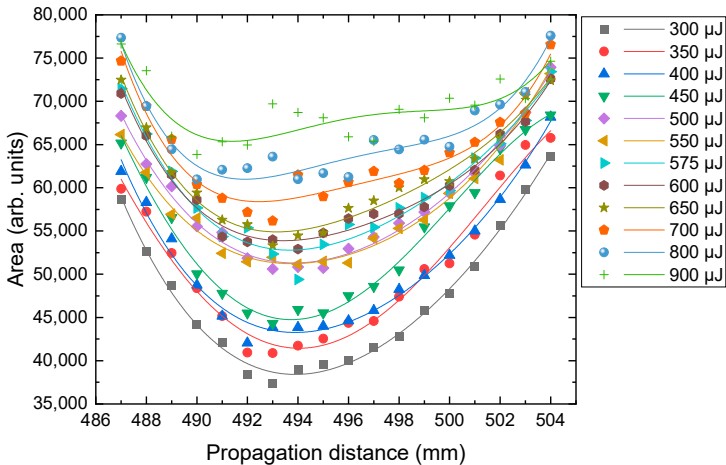

**Figure 6.** The areas of the ablated spots by vortex beam with *m* = 3 for various input energies. The points are the experimental obtained spots areas, and the lines are corresponding polynomial fittings.

Additionally, instead of using metal samples such as aluminum or copper plates, we used photosensitive paper for laser ablation in consideration of the challenge to obtain ablation patterns under relatively low input laser energy conditions. Although this choice

would result in larger beam ablation spots due to the relatively low ablation threshold of the photosensitive paper, it did not hinder us from obtaining the evolution of the beam size along the laser propagation direction for various input laser energies. However, the plots shown in Figures 4–6 lack clear indications of filamentation formation; even a filamentation regime might have been achieved under certain conditions. The filamentation should be submerged in the ablation spots during the filamentation process. This limitation arises from the use of photosensitive paper.

It is also worth noting that the measured critical power value is experiment-dependent. First, the choice of a suitable measurement technique is crucial, as different methods are based on different physical mechanisms. Consequently, there are variations in accuracy and sensitivity, resulting in different measurement outcomes. Therefore, using an appropriate experimental method to determine the critical power is of great importance. For instance, the approaches of P-scan, S-scan, and PMT fluorescence measurement typically exhibit higher sensitivity compared to the focal-shifting method [33,39,48]. Furthermore, the determination of the critical power for self-focusing is also influenced by experimental conditions, particularly the laser pulse duration and the external focusing conditions [29,30,39,49–51].

## 4. Conclusions

The critical power for self-focusing of femtosecond laser pulses in air is experimentally determined by evaluating the focal spots. This method relies on the fact that when the laser power exceeds a certain power, the focal position will move toward the laser source, while, below the critical power, the focal position remains unchanged. By evaluating the ablated spot area at various propagation distances for different input energies, the displacement of the focal position as a function of input laser energy is obtained. Consequently, the critical powers for the self-focusing of femtosecond Gaussian and vortex beams with $m = 1$ and 2 are successfully determined, respectively. However, our results also indicate that this method, along with the commonly used focus-shifting method, is not suitable for higher-order vortex beams.

One crucial aspect to consider is whether the experimental measured crossover power is suitable to be regarded as the critical power for self-focusing of femtosecond pules [27,29–32,34,39,48,49,52,53]. The theoretical definition of the critical power for self-focusing of a cw laser is well established. However, determining the critical power in the ultrashort-pule case is still a significant challenge. Furthermore, Polynkin and Kolesik concluded that the critical power concept is not straightforwardly applicable to the ultrashort-pulse case, and no particular value of peak pulse power can be viewed as a sharp demarcation line between linear and nonlinear propagation regimes [53]. However, some experimental studies demonstrated that as the input laser energy increases, the laser pulse experiences different nonlinear propagation regimes. As reported by Akturk et al. in Ref. [33], prior to the formation of filament, there exists a region (termed region II in the reference) where the self-focusing effect starts to play a significant role and the nonlinear focus moves towards the laser. In this region, the laser focus starts to move, which is the physical basis of the focal-shifting method proposed by Liu et al. in Ref. [27] for experimentally determining the critical power for self-focusing. Therefore, considering the characteristic observed in their data plot, the power at the onset of region II can be regarded as the critical power for self-focusing. Furthermore, region III represents a filamentation-dominated region, and thus the power at the beginning of this region can be regarded as the critical power for filamentation. Interestingly, in our recent work [48], we observed a similar transition of laser propagation regimes from self-focusing to filamentation, where we defined two critical powers: one for self-focusing and the other for mature filamentation. Hence, we contend that using just one critical power is inadequate for comprehensively studying the filamentation process in certain cases [33,35,36,48], and it is also reasonable to assign the crossover power as the self-focusing critical power at which the self-focusing effects become evident, which has been used in our recent work and other relevant studies [27,30–32,34,39,48,49,52]. These experimental measurements of critical

powers under various conditions by using different methods are of great importance in the field of femtosecond filamentation. These measurements provide valuable reference values, in addition to the nominal ones, that are essential for both experimental investigations and practical applications of femtosecond filamentation.

Furthermore, the $N_2$ and $O_2$ molecules in excited states or thermal waveguide structures induced by laser pulses may have an impact on the subsequent pulses [12,54], and also on the determination of critical power for self-focusing. The complex dynamics can be revealed through the manipulation of the laser repetition frequency, which requires further studies.

**Author Contributions:** Conceptualization, methodology, and project administration, L.Z. and Z.H.; investigation, H.S., B.Y., F.L., D.L. and J.C.; data curation, visualization and writing—original draft preparation, H.S.; supervision and writing—review and editing, L.Z.; resources and funding acquisition, Z.H. and Y.C. All authors have read and agreed to the published version of the manuscript.

**Funding:** This research was funded by the National Natural Science Foundation of China (12074228, 12204282, and 12204284), Natural Science Foundation of Shandong Province (ZR2021MA023), Taishan Scholar Project of Shandong Province (tsqn201812043 and tsqnz20221132), and Innovation Group of Jinan (2020GXRC039).

**Institutional Review Board Statement:** Not applicable.

**Informed Consent Statement:** Not applicable.

**Data Availability Statement:** The data underlying the results presented in this paper are not publicly available at this time but may be obtained from the authors upon reasonable request.

**Conflicts of Interest:** The authors declare no conflicts of interest.

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
