# Peer review of "An Experimental Determination of Critical Power for Self-Focusing of Femtosecond Pulses in Air Using Focal-Spot Measurements"

_photonics, doi:10.3390/photonics11010066_

Round 1

Reviewer 1 Report

Comments and Suggestions for Authors

This review evaluates the manuscript by Song et al., titled "Experimental determination of critical power for self-focusing of femtosecond pulses in air by focal-spot measurement." Overall, the paper is well-written, presents high-quality experimental work, and, although not particularly groundbreaking, it deserves consideration for publication in Photonics. Nevertheless, there is a significant flaw that must be thoroughly addressed in a revision. Additionally, several minor issues and typographical errors need correction.

***Main flaw (requiring attention in the revision)***

The manuscript claims that authors' method determines the critical power (P_cr) for self-focusing. However, I believe the value derived from their measurements is not the actual P_cr but a somewhat lower value where self-focusing begins to manifest, yet is insufficient to form a filament. Specifically:

1) The authors assert, following an argument from Reference 5, that the peak power corresponding to the initial measurable shift of the beam's waist is P_cr. However, a subsequent study (Reference 10) presents experimental data (Figure 2) contradicting this, showing that the shift in focus position starts at approximately 0.5*P_cr.

2) The experimental setup involves a refractive focusing lens. The nonlinear refraction in this lens could potentially cause the observed minor shift in focus at high pulse energies. The authors should provide an estimation of the lens's B-integral to confirm that its contribution to beam propagation is negligible.

3) The plots showing the areas of the ablated spot versus the position along the beam waist lack clear indications of filament formation, suggesting that the filamentation regime might not have been achieved. The authors need to address this.

***Minor issues for correction***

1) In Section 2 "Experimental setup," please include details about the laser repetition rate and the number of pulses accumulated in each exposure of the burn paper.

2) Provide a brief explanation of the logic behind the "Wand tool" used in the software package for image analysis.

3) The "yellow line" mentioned in the text on line 118 is nearly invisible in Fig. 2(a). Consider adjusting its color for better visibility.

4) On line 122, replace "Focal lens" with "Focusing lens" for accuracy.

Author Response

Please find the response in the uploaded file.

Reviewer 2 Report

Comments and Suggestions for Authors

Manuscript under review is very interesting and suitable for publication on Photonics journal. Authors measured critical power of femtosecond pulses self-focusing. But current version is partly unclear and extremely short, so I recommend to add some necessary physics. Please, see remarks below.

1.            Experimental setup contains polarizer, half-wave plate, focusing lens, quarte-wave plates, vortex wave plate, hence, influence of these elements on pulse duration should be discussed. Group velocity dispersions for materials of these elements should be presented in text.

2.            For 1 kHz repetition rate, density of metastable level N2(A 3S+u) (nitrogen molecule), which is created by 1 pulse, is not zero for second, third and subsequent pulses. Therefore, plasma will be created earlier for these pulses. How does repetition rate influence on measurement of critical power (please, describe using dynamics of excited state density for N2 and O2 molecules)?

3.            How were 30 pulses selected for 1 kHz repetition rate (ablation pattern consists of 30 pulses)?

4.            “During the analysis, we maintain the same tolerance of 30 in the Wand Tool of the software”. What does tolerance 30 mean?

5.            Typical intensity distribution should be added to figure 2. Difference between pulses with m=1,2,3 is unclear.

6.            How were ablation area measured? Please, mark it by visible curves on each figure and describe in detail. “For instance, the selected area for a typical Gaussian beam pattern is marked by a yellow line in the Figure 2(a)”. There is no yellow line in figure.

7.            Peak pulse intensity estimation based on ablation area will be useful.

Author Response

(The authors gave the same response as above.)

Reviewer 3 Report

Comments and Suggestions for Authors

Comments:

In the manuscript titled “Experimental determination of critical power for self-focusing of femtosecond pulses in air by focal-spot measurement,” the authors have proposed an experimental approach to investigate the critical power for self-focusing of both femtosecond Gaussian and vortex beams with relatively low topological charges by analyzing the changes in the focal spot at different propagation distances. To support this, the authors presented experimental data and discussion; however, I think the manuscript has the following significant issues.

1.     Line 39, the authors talk about only critical powers, may want to remove the word ‘respectively’.

2.     No ref. or manufacturer details about the Ti: sapphire chirped pulse amplification system. What is the output power?

3.     Line 86, the authors should provide details of the microscope or ref. How did the authors reduce the power at the focal before inserting the detector at different positions? Wasn’t it saturated?

4.     How did the singularities of m=2, 3 vortex beams look at the focal position? Is it similar to m=1? The authors should provide the images at least in the supporting.

5.     The authors should provide a reference for ImageJ software.

6.     For the Gaussian beam, the area near the focal point looks less distributed compared to away from it, at different powers. However, it is more diffused in the case of vortex beams at focus as compared to away. Why is that so?

7.     The authors talked about the shift in focal position with propagation. However, the intensity/fluence is dependent on beam diameter too. Could the authors explain it?

Overall, I think the article is concise and easy to understand. However, addressing these points would enhance the clarity and quality of the manuscript.

Comments on the Quality of English Language

Minor editing is needed.

Author Response

(The authors gave the same response as above.)

Round 2

Reviewer 1 Report

Comments and Suggestions for Authors

The authors have satisfactorily addressed all the issues raised in my initial review, except for the most crucial one. This key issue must be resolved before the manuscript is suitable for publication.

To reiterate, the optical power measured and labeled as the "critical power of self-focusing" (Pcr) by the authors does not represent the true Pcr. The authors correctly define Pcr as the power where self-focusing overtakes diffraction divergence, eventually leading to beam collapse and filament formation. However, the manuscript's measurements identify the power where self-focusing first becomes detectable, influenced by the method's sensitivity. Figure 2 in Ref 10, which I attach for reference, illustrates this difference. The power reported in the current manuscript corresponds to the onset of detectable self-focusing (akin to zone II in Fig 2 of Ref 10), rather than the true beginning of self-focusing dominated propagation (zone III in the figure).

figure: Fig. 2

Two aspects of the authors' response are concerning:

1.     The claim that measurement results may vary with experimental conditions raises a red flag. While Pcr indeed depends on the laser pulse's spatio-temporal structure, its variation with measurement methods and setup configuration should be scrutinized. This variation implies that the measured power may represent the limit of self-focusing detection, which can indeed differ based on the method's sensitivity and setup specifics.

2.      The authors' justification, based on three self-citations in the same journal over a short period, lacks external validation. While publication in a peer-reviewed journal is a strong indicator of validity, it does not preclude the possibility of errors. As researchers, we must identify and halt the propagation of such errors.

In summary, the manuscript represents a quality research effort suitable for Photonics. However, the measured optical power should not be termed Pcr, and its experiment-dependent nature must be clarified. I suggest renaming the manuscript to "Experimental Characterization of Self-Focusing..." and changing the notation for the measured power to something like "self-focusing detection limit, Psf-det." It would be commendable if the authors acknowledge any misinterpretation in their previous work and include a statement in this manuscript to prevent future misunderstandings.

Author Response

Please find the our Responses to Reviewer's Comments in the attached file.

Reviewer 2 Report

Comments and Suggestions for Authors

Article can be published

Author Response

Comments and Suggestions for Authors: Article can be published Response: We would like to express our gratitude to Reviewer 2 for providing valuable comments and insightful suggestions on our manuscript.

Round 3

Reviewer 1 Report

Comments and Suggestions for Authors

Dear Editor,

I regret to say that I don't see this discussion converging to a satisfactory conclusion. The author's argument that the definition of Pcr can be presented as an open question, in my opinion, is not acceptable. Pcr has a well-defined physical meaning and leaves no room for interpretation. The focus-shift method introduced in 2005 in Ref. 5, and adapted by the authors, measures a power that is different from Pcr. Interpreting it as such was unambiguously declared incorrect in a 2011 publication (Ref. 7), which, in the current revision, is added to the reference list twice, also as Ref. 24. Ref. 7/24 defines the threshold power associated with the onset of focus movement as “crossover power” (Pc−o) and demonstrates that it is not equal to Pcr. Here is an excerpt from Ref. 7/24:

“Indeed, the main result of this study is that the crossover power as observed in a nonlinear focus-shift experiment cannot be directly identified with the nominal critical power for self-focusing collapse. In general, Pc−o is significantly affected by defocusing caused by free electrons, and it also depends on the focusing geometry.”

Considering this, I cannot recommend the manuscript under review for publication in Photonics in its current form.
